# Nationwide Real-World Data of Microsatellite Instability and/or Mismatch Repair Deficiency in Cancer: Prevalence and Testing Patterns

**DOI:** 10.3390/diagnostics14111076

**Published:** 2024-05-22

**Authors:** Elena Fountzilas, Theofanis Papadopoulos, Eirini Papadopoulou, Cedric Gouedard, Helen P. Kourea, Pantelis Constantoulakis, Christina Magkou, Maria Sfakianaki, Vassiliki Kotoula, Dimitra Bantouna, Georgia Raptou, Angelica A. Saetta, Georgia Christopoulou, Dimitris Hatzibougias, Electra Michalopoulou-Manoloutsiou, Eleni Siatra, Eleftherios Eleftheriadis, Evangelia Kavoura, Loukas Kaklamanis, Antigoni Sourla, George Papaxoinis, Kitty Pavlakis, Prodromos Hytiroglou, Christina Vourlakou, Petroula Arapantoni-Dadioti, Samuel Murray, George Nasioulas, Grigorios Timologos, George Fountzilas, Zacharenia Saridaki

**Affiliations:** 1Department of Medical Oncology, St. Lukes’s Clinic, 55236 Thessaloniki, Greece; 2Medical Oncology, European University Cyprus, 2404 Nicosia, Cyprus; 3Molecular Diagnostics Laboratory, KARYO Ltd., 54622 Thessaloniki, Greece; papadopoulos@karyo.gr (T.P.); timologos@karyo.gr (G.T.); 4Genekor M.S.A., 15344 Athens, Greece; eirinipapad@genekor.com (E.P.); gnasioulas@genekor.com (G.N.); 5BioPath Innovations SA, 15124 Athens, Greece; cedric.gouedard@biopathinnovations.com (C.G.); smgenedb@gmail.com (S.M.); 6Department of Pathology, University Hospital of Patras, 26504 Patras, Greece; hkourea@yahoo.com (H.P.K.); dimitra.bantouna@yahoo.gr (D.B.); 7Genotypos Science Labs M.S.A., 11525 Athens, Greece; pconstantoulakis@genotypos.gr (P.C.); gchristopoulou@sciencelabs.gr (G.C.); 8Department of Pathology, Evangelismos Hospital, 10676 Athens, Greece; cmagkou@yahoo.com (C.M.); ch.vourlakou@yahoo.gr (C.V.); 9Laboratory of Translational Oncology, Medical School, University of Crete, 71410 Heraklion, Greece; mimasf19@gmail.com; 10Department of Pathology, School of Health Sciences, Faculty of Medicine, Aristotle University of Thessaloniki, 54124 Thessaloniki, Greece; vkotoula@auth.gr (V.K.); graptou@auth.gr (G.R.); phytiro@auth.gr (P.H.); 11Laboratory of Molecular Oncology, Hellenic Foundation for Cancer Research/Aristotle University of Thessaloniki, 54006 Thessaloniki, Greece; fountzil@auth.gr; 12First Department of Pathology, Medical School, National and Kapodistrian University of Athens, Laiko General Hospital, 15772 Athens, Greece; asaetta@med.uoa.gr; 13microDiagnostics LP, Private Surgical & Molecular Pathology Laboratory, 54622 Thessaloniki, Greece; dhbugias@gmail.com (D.H.); electramm@hotmail.gr (E.M.-M.); 14Department of Pathology, Henry Dunant Hospital, 11526 Athens, Greece; eleni.siatra@yahoo.gr (E.S.); p.arapantoni@dunant.gr (P.A.-D.); 15Istodierevnitiki S.A., 54622 Thessaloniki, Greece; drleft@biopsy.gr; 16Pathology Department, IASO Women’s Hospital, 15123 Athens, Greece; evangeliakavoura@gmail.com (E.K.); epavlaki@med.uoa.gr (K.P.); 17Pathology Department, Onassis Cardiac Surgery Center, 17674 Athens, Greece; loukasgka@yahoo.gr; 18Department of Pathology and Laboratory Medicine, Bioiatriki Laboratories, 11528 Athens, Greece; asourla@hotmail.com; 19Second Department of Internal Medicine, Agios Savvas Cancer Hospital, 11522 Athens, Greece; georgexoinis@gmail.com; 20Aristotle University of Thessaloniki, 54124 Thessaloniki, Greece; 21Department of Medical Oncology, German Oncology Center, 4108 Limassol, Cyprus; 22First Oncology Department, Metropolitan Hospital, 18547 Piraeus, Greece; zeniasar@gmail.com; 23Asklepios, Oncology Department, 71303 Heraklion, Greece

**Keywords:** biomarker, Greece, immunotherapy, MMR, MSI, nationwide, real-world data

## Abstract

Determination of microsatellite instability (MSI)/mismatch repair (MMR) status in cancer has several clinical implications. Our aim was to integrate MSI/MMR status from patients tested in Greece to assess the prevalence of MSI-high (MSI-H)/deficient MMR (dMMR) per tumor type, testing patterns over time and concordance between MSI and MMR status. We retrospectively recorded MSI/MMR testing data of patients with diverse tumor types performed in pathology and molecular diagnostics laboratories across Greece. Overall, 18 of 22 pathology and/or molecular diagnostics laboratories accepted our invitation to participate. In the 18 laboratories located across the country, 7916 tumor samples were evaluated for MSI/MMR status. MSI/MMR testing significantly increased in patients with colorectal cancer (CRC) and other tumor types overtime (*p* < 0.05). The highest prevalence was reported in endometrial cancer (47 of 225 patients, 20.9%). MSI-H/dMMR was observed in most tumor types, even in low proportions. Among 904 tumors assessed both for MSI and MMR status, 21 had discordant results (overall discordance rate, 2.3%). We reported MSI-H/dMMR prevalence rates in patients with diverse cancers, while demonstrating increasing referral patterns from medical oncologists in the country overtime. The anticipated high rate of concordance between MSI and MMR status in paired analysis was confirmed.

## 1. Introduction

Mismatch repair (MMR) proteins are responsible for excising DNA mismatches introduced by DNA polymerase during cell division, commonly occurring in repetitive DNA sequences (known as microsatellites). Impairment of the MMR system leads to microsatellite instability (MSI), which is characterized by the accumulation of mismatches in repeated sequences (1). Defects in the MMR system can be assessed using polymerase chain reaction (PCR)-based assays testing for MSI, next-generation sequencing (NGS) or immunohistochemical (IHC) analysis of MMR protein expression, including MLH1, MSH2, MSH6 and PMS2 proteins (2).

Determination of MSI/MMR status in cancer has several clinical implications. First, MMR status has been shown to provide valuable prognostic information, mainly in patients with colorectal cancer (CRC) [1,2,3]. Additionally, it has been demonstrated that MSI/MMR status may be associated with lack of benefit from adjuvant chemotherapy in stage II CRC [4,5]. Second, MSI and/or MMR-deficiency (dMMR) is being currently used as a biomarker, predicting response to checkpoint inhibitors as it correlates with significant clinical benefit across tumor types [6,7,8,9]. Specifically, approved immune checkpoint inhibitors for the treatment of patients with MSI-H/dMMR tumors include pembrolizumab or dostarlimab, administered alone as monotherapies, and nivolumab with ipilimumab, administered as combination therapy. Importantly, MSI-H and dMMR may be associated with germline pathogenic variants in the respective MMR genes, the presence of which needs to be investigated to identify patients with Lynch syndrome [10].

MSI-H/dMMR has been reported in diverse tumor types [11]. Tumor types more commonly found to be dMMR are endometrial, colorectal and gastric cancers [11,12]. The high prevalence of dMMR in colorectal and endometrial tumors, in combination with the significant clinical impact of MMR status to the patient’s management, underscores the importance of screening for dMMR. In addition, due to the benefit derived from treatment with checkpoint inhibitors in patients with MSI-H/dMMR tumors, universal testing is currently recommended for all patients with advanced cancer.

In Greece, MSI/MMR testing is reimbursed only in patients with CRC, irrespective of the age of diagnosis. In patients with any other tumor type, testing is performed at out-of-pocket costs, if affordable by the patient. On the contrary, immune checkpoint inhibitors are reimbursed for all patients with advanced MSI-H/dMMR tumors, regardless of the primary site. Due to the lack of national cancer registries, the frequency of MSI-H/dMMR in patients with diverse tumor types is not known. Our aim was to integrate MSI and/or MMR status from patients tested nationwide to assess the prevalence of MSI-H/dMMR per tumor type in patients in Greece. In addition, we evaluated the concordance between MSI-H and dMMR, if both tests were performed in the same tumor sample.

## 2. Patients and Methods

### 2.1. Participating Centers/Laboratories

We retrospectively recorded data on MSI/MMR testing of patients with diverse tumor types performed in different laboratories across Greece. The head of each pathology or molecular diagnostics laboratory, known to perform either MSI testing and/or MMR IHC analysis, was contacted by the principal investigator. Laboratories, from both the private and public sector, were invited to participate in this project. An official letter of intent was sent to the Hellenic Society of Pathology to notify laboratories about the project and invite their participation. Laboratories of molecular diagnostics were identified through contacting the members of the Hellenic Cooperative Oncology Group and using search engines.

### 2.2. Data Collection

Data on MSI/MMR status were recorded along with clinical and histological parameters (tumor type, histology, stage, grade) when available. Importantly, the date of testing was recorded in order to assess the possible increase in referral patterns over time. All data were anonymized and provided with a unique identification number. A confidentiality agreement between each laboratory and the principal investigator was signed to preserve privacy issues of each lab. A waiver of consent was obtained from the scientific committee of General Hospital “Agioi Anargiri” (108/30-1-2020), since this research presented no risk of harm to participants and involved no procedures for which written consent is normally required.

### 2.3. Statistical Analysis

Τhe primary endpoint of the study was to assess the prevalence of MSI and/or MMR status in diverse tumor types. Secondary endpoints included referral rates overtime and the rate of discordance between MSI and MMR IHC assessments. Categorical variables were presented using frequencies and percentages, whilst continuous variables were presented using median alongside the interquartile range (IQR). Spearman’s correlation was run to determine the association between two continuous variables. The Mann–Whitney test was used to examine differences in the distribution of continuous variables between groups and the chi-square test to assess whether there was a statistically significant relationship between categorical variables. Significance level was set to 0.05.

## 3. Results

### 3.1. MSI/MMR Prevalence

Of 22 pathology and/or molecular diagnostics laboratories that were contacted, 18 confirmed their participation. MSI testing only was performed in eight laboratories, while MMR IHC testing only was performed in six laboratories. Four laboratories provided results for both MSI and MMR status. MSI testing was performed either by PCR or NGS depending on the physician’s referral, respectively (Figure 1, consort diagram). Details on different methodologies are included in Appendix A.

From March 1999 to April 2022, 7916 tumor samples were evaluated for MMR and/or MSI status in those centers. MSI-H and/or dMMR were identified in 838 (10.6%) samples. The highest prevalence was reported in patients with endometrial cancer (47 of 225 patients, 20.9%). The prevalence of MSI-H and/or dMMR in other tumor types is presented in Table 1. An unexpectedly high prevalence of MSI-H/dMMR tumors was observed in selected tumor types, possibly reflecting referral patterns due to high clinical suspicion.

### 3.2. Referring Patterns

A statistically significant increase in MSI/MMR testing was observed in patients with colorectal cancer overtime: 91 (≤2012), 71 (2013–2014), 283 (2015–2016), 1475 (2017–2018) and 1662 tumors (2019–2020) (*p* < 0.05). Similarly increased referral numbers were observed in patients with other tumor types. Referrals in association with time are shown in Figure 2. The graph shows a significant increase in MSI/MMR testing, both in patients with colorectal cancer and with other tumor types.

### 3.3. MSI and MMR Status Concordance

Among 7916 samples, 904 were assessed for both MSI and MMR status; 21 had discordant results for MSI and MMR status (overall discordance rate of 2.3%). Specifically, among the MSI-H tumor samples (*n* =102), 8 tumors were pMMR, while among the MSS tumor samples (*n* = 802), 13 tumors were shown to be dMMR. Discordant cases were not reviewed to assess whether discordance was due to IHC misclassification or MSI testing errors.

## 4. Discussion

This is the first study to report referral patterns and incidence of MSI-H/dMMR in Greek patients with diverse tumor types. MSI/MMR status of tumor samples assessed in laboratories of pathology and/or molecular diagnostics laboratories nationwide were combined to demonstrate that MSI-H/dMMR endometrial and colorectal cancer represented a significant proportion of referrals, possibly due to the increased probability of MSI-H/dMMR in such cancers and the recommendation of universal MIS/MMR testing in those tumor types. Additionally, MSI-H/dMMR characterized diverse tumor types, even in low rates, thus underlying the need to assess MSI/MMR status in all patients with advanced cancer who might benefit from this approach. Importantly, our study confirmed the high degree of concordance between MSI and MMR status in paired analysis.

In our study, a low rate of discordance between MMR and MSI status in analysis of paired samples was reported. Similarly, other investigators have reported discordant rates, ranging from 1.6% to 9.1% in patients with colorectal (15–17) or diverse tumor types (18). Differences between studies might be associated with time of testing, testing techniques, testing methodologies (assessment of two compared with four MMR proteins) and scientific expertise (19). We did not review discordant cases to assess whether discordance was due to IHC misclassification or MSI testing errors. However, we need to stress that the 13 cases that might have been wrongly classified may have profound implications for the patients in terms of treatment (immunotherapy) and screening methodology for the presence of Lynch syndrome. Therefore, in clinical practice in cases where there is high clinical suspicion, IHC may be worth repeating or accompanied by MSI testing by PCR. Since 5–10% of tumors in Lynch syndrome are not MSI-H or dMMR [13,14] and since there is a 5–10% false-negative rate with IHC testing [15,16], in cases where personal or family history suggestive of Lynch syndrome (i.e., multiple Lynch syndrome-related tumors in a family), international guidelines suggest the performance of genetic testing in those individuals.

In one study, reevaluation of 51 discordant cases demonstrated that only 12 paired samples remained discordant after reclassification (discordance rate 0.4%), underlying the importance of an expert team evaluating samples and performing the techniques [17]. Improving testing techniques and providing accurate results is critical, since false negative results might prohibit the use of innovative and efficacious treatments, including immunotherapy, in the respective patients or incorrectly alter their treatment plan.

MSI/MMR status is a robust biomarker for the use of immunotherapy in patients with advanced cancers. Immune checkpoint inhibitors have been approved for the treatment of patients with advanced MSI-H/dMMR cancers irrespective of tumor histology and origin [7,8,9]. These data underscore the clinical significance of this biomarker being tested in all patients with advanced cancer, as defined by national and international guidelines. Furthermore, recent data support the use of neoadjuvant immunotherapy in earlier stages of MSI-H/dMMR CRC [18]. Our study demonstrated that MSI/MMR testing referrals increased overtime, possibly due to biomarker reimbursement along with increasing evidence regarding the benefit of immunotherapy in MSI-H/dMMR tumors. Specifically, MSI testing reimbursement was initiated in late 2014. However, MSI testing referrals in our study dramatically increased during the years 2017/2018, possibly due to pembrolizumab FDA approval for the treatment of MSI-H/dMMR tumors in 2017. Finally, renewed guidelines underlying universal screening in colorectal and endometrial tumors may also have played a role in the increase in MSI testing.

Despite the preventive, prognostic and predictive clinical value of the MSI/MMR as a biomarker, in Greece it is still only reimbursed in CRC patients, whereas the administration of checkpoint inhibitors to patients with MSI-H/dMMR advanced tumors is reimbursed for all patients with national insurance. Studies have shown that universal tumor testing strategies are cost-effective in patients with CRC and endometrial cancer for the aforementioned reasons [19,20]. Whether testing all patients with advanced tumors for MSI-H/dMMR should become routine clinical practice needs to be considered. Real-world data on the prevalence and clinical actionability of MSI-H/dMMR testing can provide evidence for improved planning and coordination policies under national health systems worldwide [21].

Limitations of our study include its retrospective nature; the lack of detailed clinicopathological, treatment and outcome data; and the absence of germline data in patients with MSI-H/dMMR tumors. In addition, the size of the cohort with both MSI/MMR status limits the strength of the comparison results. Finally, a major limitation of our study is the lack of proportions of individual tumor types that were processed by collaborating laboratories compared with the tumors not referred for testing. These data are even more relevant in recent years, where recommended guidelines for MSI/MMR testing in clinical practice have been established. Missing data, a common phenomenon often observed in real-world studies, is a major limitation that might lead to gaps or biases in the analysis.

Strengths of our study include its multicentric nature with the participation of both pathology and molecular diagnostics laboratories and the nationwide collection of MSI/MMR status in diverse tumor types, thus ensuring representation of the Greek population as a whole. Furthermore, the concordance between dMMR and MSI-H in diverse tumor types was evaluated in our study, which is critical in daily clinical practice.

## 5. Conclusions

In conclusion, we reported real-world data on referral testing practices, prevalence of MSI-H/dMMR across tumor types and concordance between MSI-H and dMMR in selected patients in Greece. Now, more than ever, in the era of precision medicine and significantly effective innovative yet costly therapeutic agents, real-world data on the current status of the use of predictive biomarkers and their clinical application can provide important evidence, especially in countries where formal registries are lacking.

## Figures and Tables

**Figure 1 diagnostics-14-01076-f001:**
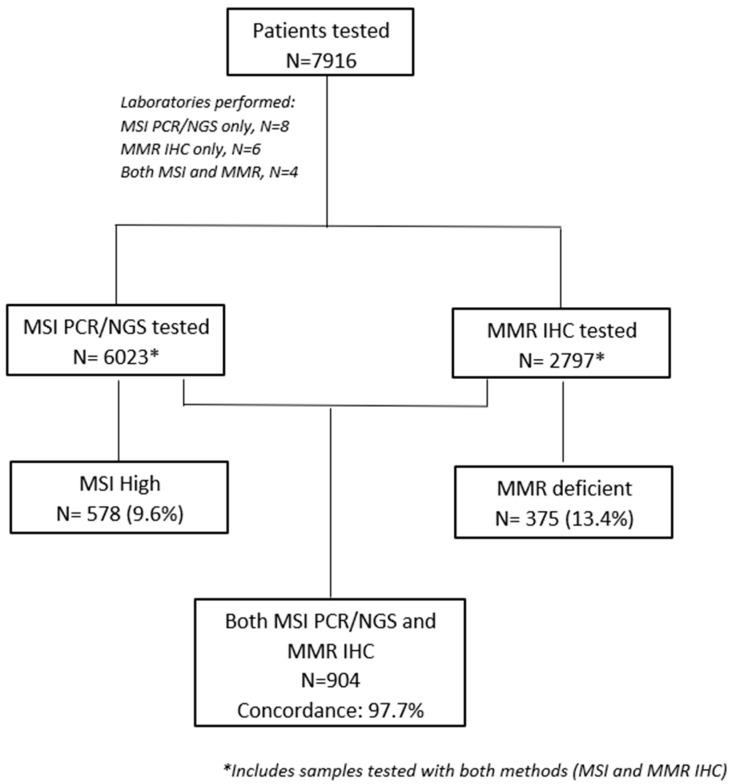
Consort diagram.

**Figure 2 diagnostics-14-01076-f002:**
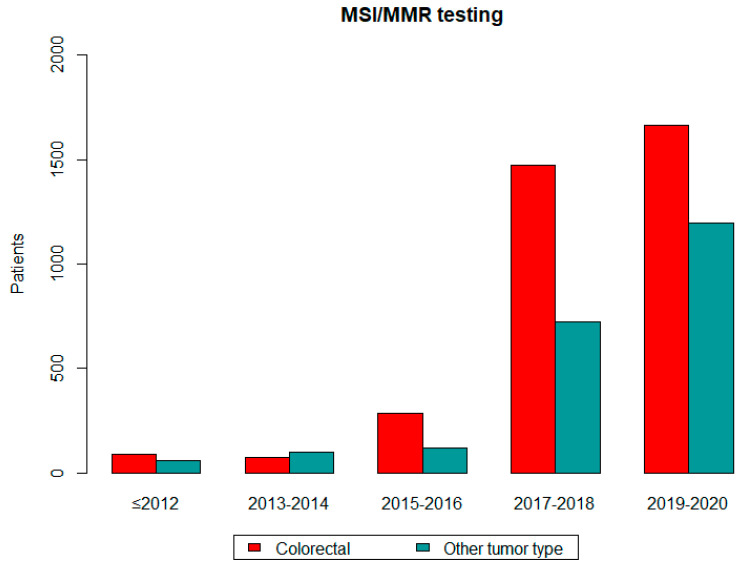
MSI/MMR testing over time.

**Table 1 diagnostics-14-01076-t001:** Patient characteristics.

Factor *	Total (*N* = 7916)	MSS and/or pMMR (*N* = 7078)	MSI-H and/or dMMR (*N* = 838)
Age at testing Median (IQR) (*N* = 5577)	66.4 (15.9)	66.3 (15.7)	67.4 (17.8)
Gender (*N* = 6639)			
Female	2967 (44.7%)	2643 (89.1%)	324 (10.9%)
Male	3672 (55.3%)	3334 (90.8%)	338 (9.2%)
Cancer type (*N* = 7820)			
Colorectal	5068 (64.8%)	4456 (87.9%)	612 (12.1%)
Endometrial	225 (2.9%)	178 (79.1%)	47 (20.9%)
Other tumor type	2527 (32.3%)	2362 (93.5%)	165 (6.5%)
Other GI **	478 (18.9%)	440 (92.1%)	38 (7.9%)
Lung	276 (10.9%)	268 (97.1%)	8 (2.9%)
Pancreas	296 (11.7%)	289 (97.6%)	7 (2.4%)
Other gynecological º	150 (5.9%)	145 (96.7%)	5 (3.3%)
Prostate	91 (3.6%)	86 (94.5%)	5 (5.5%)
Breast	93 (3.7%)	90 (96.8%)	3 (3.2%)
Liver	49 (1.9%)	45 (93.8%)	4 (8.3%)
CNS	25 (1.0%)	25 (100%)	0 (0.0%)
Genitourinary	28 (1.1%)	25 (89.3%)	3 (10.7%)
Melanoma	12 (0.5%)	12 (100%)	0 (0.0%)
HNC	13 (0.5%)	13 (100%)	0 (0.0%)
Sarcoma	16 (0.6%)	15 (93.8%)	1 (6.3%)
Adrenal Gland	2 (0.1%)	1 (50.0%)	1 (50.0%)
GIST	4 (0.2%)	4 (100%)	0 (0.0%)
Lymphoma	1 (0.04%)	1 (100%)	0 (0.0%)
Mesothelioma	4 (0.2%)	4 (100%)	0 (0.0%)
Skin	1 (0.04%)	1 (100%)	0 (0.0%)
Thymus	3 (0.1%)	2 (66.7%)	1 (33.3%)
Thyroid	7 (0.3%)	7 (100%)	0 (0.0%)
NET	1 (0.04%)	1 (100%)	0 (0.0%)
Primary site not reported	977 (38.7%)	888 (90.9%)	89 (9.1%)
Stage at diagnosis (*N* = 3216)			
I–III	2878 (89.5%)	2509 (87.2%)	369 (12.8%)
IV	338 (10.5%)	305 (90.2%)	33 (9.8%)

Abbreviations: CNS—central nervous system, HNC—head and neck, GI—gastrointestinal, GIST—gastrointestinal stromal tumor, dMMR—mismatch repair system deficiency, pMMR—mismatch repair system proficiency, MSS—microsatellite stable, MSI-H—high microsatellite instability, N—number, NET—neuroendocrine tumor. * Data not available for all subjects. Values presented as N (column %). Percentage in MSS/pMMR and MSI-H/dMMR columns indicates the prevalence of MSS/pMMR and MSI-H/dMMR, respectively, in each value of the variable. ** Includes gastric/gastroesophageal, small bowel, anal and biliary cancers. The majority of the tumors (80.0%) were gastric/gastroesophageal cancers, and the prevalence of MSI-H/dMMR in this group was 5.5%. º Includes ovarian, cervical, vaginal and peritoneal cancer.

## Data Availability

The data presented in this study are available in the article, while further details can be obtained on request from the corresponding author.

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
