# Peer review of "Nationwide Real-World Data of Microsatellite Instability and/or Mismatch Repair Deficiency in Cancer: Prevalence and Testing Patterns"

_diagnostics, 2024, doi:10.3390/diagnostics14111076_

Round 1

Reviewer 1 Report

Comments and Suggestions for Authors

This straightforward manuscript describes MMR/MSI testing performed in Greece irrespective of cancer stage or type. The study is of interest as this type of information is not generally available for any country. Furthermore, the data have multiple uses as a guide for treating physicians and service providers as well as a platform for public health initiatives including the funding of tests and allocation of resources. In general, the manuscript is well-structured, data have been presented clearly, and the authors have not overinterpreted results. Due to the ‘real world’ source of data, there are many limitations of the study, however, despite these shortcomings, the study is worthy of publication as a snapshot of current practices and as a basis for future more comprehensive research. The authors could consider the following comments prior to publication.

 1. A major limitation of the study is that the total numbers of individual tumour types processed by the laboratories were not collected/presented. This means that the proportion of individual tumour types (e.g. colorectal cancer) that underwent MSI/MMR testing is unknown. This omission is less relevant for years at the start of the collection period but is of interest more recently where recommended guidelines for testing and for clinical practice have been established. A comment to this effect and to the amount of missing data or gaps in data (which is inevitable when collecting information from real world practice) could be added to the Discussion.

2. In Figure 2, it is clear that the numbers of MSI/MMR tests performed was dramatically increased in 2017-2018 compared to 2015-2016. Was a potential reason for this identified? (e.g. reimbursement for MSI testing for CRC commenced, change in clinical management practices, new pathology practice(s) offering/marketing MSI testing). This could be addressed briefly in the text.

Comments on the Quality of English Language

English language and grammar are generally very good. There are minor spelling and grammatical errors and the location of some of these are listed below. In addition, there are some phrases or sentences where grammatical corrections will be required.

1. Lines 7, 8: The names of 2 of the authors are underlined. Was this intentional?

2. The details of some of the authors require revision (e.g. line 12).

3. Line 87: What does “monotherapy in combination treatment” mean?

4. Lines 100, 237: The expression is “irrespective of” (not ‘irrespectively’)

5. Line 149: Should this be ‘Cohort diagram’ (not ‘consort’)?

6. Lines 218-219: This sentence does not make sense in its current form and requires amendment.

7. Line 226: ‘Lynch syndrome-related tumors’

8. Line 241: The word ‘disease’ is redundant and can be removed.

9. Supplementary information: There are minor errors in some of these descriptions (e.g. ‘unstable’ (not ‘instable’), the % (percent) sign has not be written correctly in one of the methods, ‘quasimonomorphic’ (not ‘quacimonomorphic’), etc). This section will require checking for minor errors.

Author Response

Comments and Suggestions for Authors

REVIEWER 1

This straightforward manuscript describes MMR/MSI testing performed in Greece irrespective of cancer stage or type. The study is of interest as this type of information is not generally available for any country. Furthermore, the data have multiple uses as a guide for treating physicians and service providers as well as a platform for public health initiatives including the funding of tests and allocation of resources. In general, the manuscript is well-structured, data have been presented clearly, and the authors have not overinterpreted results. Due to the ‘real world’ source of data, there are many limitations of the study, however, despite these shortcomings, the study is worthy of publication as a snapshot of current practices and as a basis for future more comprehensive research. The authors could consider the following comments prior to publication.

  1. A major limitation of the study is that the total numbers of individual tumor types processed by the laboratories were not collected/presented. This means that the proportion of individual tumor types (e.g. colorectal cancer) that underwent MSI/MMR testing is unknown. This omission is less relevant for years at the start of the collection period but is of interest more recently where recommended guidelines for testing and for clinical practice have been established. A comment to this effect and to the amount of missing data or gaps in data (which is inevitable when collecting information from real world practice) could be added to the Discussion.

Reply: We appreciate the reviewer’s comment. We have added this major limitation to the discussion section:

“Finally, a major limitation of our study is the lack of proportions of individual tumor types that were processed by collaborating laboratories compared to the tumors not referred for testing. These data are even more relevant in recent years where recommended guidelines for MSI/MMR testing in clinical practice has been established. Missing data, a common phenomenon often observed in real-world studies, is a major limitation that might lead to gaps or biases in the analysis”.

  1. In Figure 2, it is clear that the numbers of MSI/MMR tests performed was dramatically increased in 2017-2018 compared to 2015-2016. Was a potential reason for this identified? (e.g. reimbursement for MSI testing for CRC commenced, change in clinical management practices, new pathology practice(s) offering/marketing MSI testing). This could be addressed briefly in the text.

Reply: We agree with the reviewer’s comment. We have now added the following paragraph providing potential reasons explaining the increase in MSI testing throughout the years.

“Specifically, MSI testing reimbursement was initiated in late 2014. However, MSI testing referrals in our study dramatically increased during the years 2017/2018 possibly due to pembrolizumab FDA approval for the treatment of MSI-H/dMMR tumors in 2017. Finally, renewed guidelines underlying universal screening in colorectal and endometrial tumors may also have played a role in the increase of MSI testing”.   

Comments on the Quality of English Language

English language and grammar are generally very good. There are minor spelling and grammatical errors and the location of some of these are listed below. In addition, there are some phrases or sentences where grammatical corrections will be required.

  1. Lines 7, 8: The names of 2 of the authors are underlined. Was this intentional?

Reply: We appreciate the reviewer’s comment. The names were underlined because there was a hyperlink leading to their publications. We have now removed the hyperlink to avoid any misperceptions.

  1. The details of some of the authors require revision (e.g. line 12).

Reply: Per the reviewer’s comment we have revised details in the author affiliations.

  1. Line 87: What does “monotherapy in combination treatment” mean?

Reply: We thank the reviewer for the comment. There are various immune checkpoint inhibitors approved for patients with MSI high tumors. These agents can be administered alone, as monotherapy (for instance pembrolizumab or dostarlimab), while others can be administered together (for instance combination of nivolumab and ipilimumab. To clarify this point we have revised the text as follows:

“Second, MSI and/or MMR-deficiency (dMMR) is being currently used as a biomarker, predicting response to checkpoint inhibitors, as monotherapy in combination treatment, as it correlates with significant clinical benefit across tumor types (8-11). Specifically, approved immune checkpoint inhibitors for the treatment of patients with MSI-H/dMMR tumors include pembrolizumab or dostarlimab, administered alone as monotherapies, and nivolumab with ipilimumab administered as combination therapy” (page XX, paragraph XX).

  1. Lines 100, 237: The expression is “irrespective of” (not ‘irrespectively’)

Reply: We have revised the respective sections.

  1. Line 149: Should this be ‘Cohort diagram’ (not ‘consort’)?

Reply: We thank the reviewer for the comment. Per Journal’s guidelines a completed CONSORT flow diagram is required.

  1. Lines 218-219: This sentence does not make sense in its current form and requires amendment.

Reply: We appreciate the reviewer’s suggestion. We have corrected the following sentence “However, we need to stress out that the 13 cases that might have been wrongly classified, may have profound implications for the patients in terms of treatment (immunotherapy) and screening methodology for the presence of Lynch syndrome”.

  1. Line 226: ‘Lynch syndrome-related tumors’

Reply: We have revised the sentence per the reviewer’s suggestion.

  1. Line 241: The word ‘disease’ is redundant and can be removed.

Reply: The word “disease” has been removed.

  1. Supplementary information: There are minor errors in some of these descriptions (e.g. ‘unstable’ (not ‘instable’), the % (percent) sign has not be written correctly in one of the methods, ‘quasimonomorphic’ (not ‘quacimonomorphic’), etc). This section will require checking for minor errors.

Reply: We thank the reviewer for the assessment. We have corrected the errors in the supplement.

Reviewer 2 Report

Comments and Suggestions for Authors

The manuscript “Nationwide Real-World Data of Microsatellite Instability and/or Mismatch Repair Deficiency in cancer: prevalence and testing patterns”, submitted as a communication, represents an original multicentric study. Authors have described the results of microsatellite instability/ mismatch repair status testing in different malignant tumours, collecting the data from different laboratories in Greece.

In my opinion, the major weakness of the current study is the lack of verification of the results in order to exclude errors in testing and/or evaluation of the cases. However, authors have noted that unverified data represent the real situation with microsatellite instability/ mismatch repair testing. Further, the strong and weak sides of the study have been explicitly discussed in the manuscript. Thus, I believe the presented information still can be useful for the medical society.

There are only few remarks for potential improvements:

1) Please, check the formatting of references. It should be in agreement with the “Instructions for Authors”.

2) Although the level of English language is generally high, the whole manuscript should be checked for misspells and minor grammatical errors (e.g., line 219).

Finally, I would like to thank the authors for their contribution.

Comments on the Quality of English Language

Although the level of English language is generally high, the whole manuscript should be checked for misspells and minor grammatical errors (e.g., line 219).

Author Response

Comments and Suggestions for Authors

REVIEWER 2

The manuscript “Nationwide Real-World Data of Microsatellite Instability and/or Mismatch Repair Deficiency in cancer: prevalence and testing patterns”, submitted as a communication, represents an original multicentric study. Authors have described the results of microsatellite instability/ mismatch repair status testing in different malignant tumours, collecting the data from different laboratories in Greece.

In my opinion, the major weakness of the current study is the lack of verification of the results in order to exclude errors in testing and/or evaluation of the cases. However, authors have noted that unverified data represent the real situation with microsatellite instability/ mismatch repair testing. Further, the strong and weak sides of the study have been explicitly discussed in the manuscript. Thus, I believe the presented information still can be useful for the medical society.

There are only few remarks for potential improvements:

1) Please, check the formatting of references. It should be in agreement with the “Instructions for Authors”.

Reply: We would like to thank the reviewer for the comment. We have updated the references to match the Journal’s requirements.

2) Although the level of English language is generally high, the whole manuscript should be checked for misspells and minor grammatical errors (e.g., line 219).

Reply: We appreciate the reviewer’s suggestion. We have carefully checked the manuscript for grammatical and spelling errors. We have also corrected the following sentence “However, we need to stress out that the 13 cases that might have been wrongly classified, may have profound implications for the patients in terms of treatment (immunotherapy) and screening methodology for the presence of Lynch syndrome”.
